# Small heat shock proteins determine synapse number and neuronal activity during development

**Elena Santana**◉, **Teresa de los Reyes**◉, **Sergio Casas-Tintó**◉ *

Instituto Cajal, CSIC, Madrid, Spain

◉ These authors contributed equally to this work.
* scasas@cajal.csic.es

**Data Availability Statement:** All relevant data are within the paper and its Supporting Information files.

**Funding:** We would like to declare that this research has been funded by grant BFU2015-

## Abstract

Environmental changes cause stress, Reactive Oxygen Species and unfolded protein accumulation which hamper synaptic activity and trigger cell death. Heat shock proteins (HSPs) assist protein refolding to maintain proteostasis and cellular integrity. Mechanisms regulating the activity of HSPs include transcription factors and posttranslational modifications that ensure a rapid response. HSPs preserve synaptic function in the nervous system upon environmental insults or pathological factors and contribute to the coupling between environmental cues and neuron control of development. We have performed a biased screening in *Drosophila melanogaster* searching for synaptogenic modulators among HSPs during development. We explore the role of two small-HSPs (sHSPs), sHSP23 and sHSP26 in synaptogenesis and neuronal activity. Both sHSPs immunoprecipitate together and the equilibrium between both chaperones is required for neuronal development and activity. The molecular mechanism controlling HSP23 and HSP26 accumulation in neurons relies on a novel gene (*CG1561*), which we name *Pinkman* (*pkm*). We propose that sHSPs and Pkm are targets to modulate the impact of stress in neurons and to prevent synapse loss.

## Introduction

Synaptic dynamics remodel neuronal circuits under stress conditions [1]. The Heat Shock Protein family (HSPs) is involved in preserving cellular functions such as stress tolerance, protein folding and degradation, cytoskeleton integrity, cell cycle and cell death [2–7]. HSPs are molecular chaperones that represent an intracellular protein quality system to maintain cellular protein homeostasis, preventing aggregation and promoting protein de novo folding or refolding and degradation of misfolded proteins [8]. In addition, HSPs participate in developmental functions in a stress-independent manner [9, 10]. In *Drosophila* development *small Heat Shock proteins* (*sHsps*) have a specific temporal and spatial pattern of expression [10]. In particular, *sHsp23* and *sHsp26* show high expression levels in CNS during development, suggesting a role in neural development [10].

65685P from the Spanish MICINN. The funders
had no role in study design, data collection and
analysis, decision to publish, or preparation of the
manuscript.

**Competing interests:** The authors have declared
that no competing interests exist.

sHSPs include a large group of proteins represented in all kingdoms of life [11], with a conserved protein binding domain of approximately 80 amino-acid alpha crystallin [12]. These molecular chaperones were initially described as low molecular weight chaperones that associate early with misfolded proteins and facilitate refolding or degradation by other chaperones and co-factors [11] [13]. However, members of the sHSPs have diverse functions beyond the chaperon activity including cytoskeleton assembly [14], the suppression of reactive oxygen species, anti-inflammatory, autophagy, anti-apoptotic and developmental functions (reviewed in [2]). sHSPs represent the most extended subfamily of HSPs, albeit the less conserved [15]. sHSPs have a conserved primary structure divided in three elements required for their function: 1) a variable N-terminal long-sequence related to oligomerization, 2) the conserved α-crystallin domain required for dimmers formation that represents the main hallmark of sHsps family, and 3) a flexible short C-terminal sequence mediating oligomers stability [11, 16]. Post-translational modifications in sHSPs shift the folding/degradation balance and, in consequence, alter dimer or oligomer formation and function [11, 17]. This chaperone control system modulates critical decisions for the folding or degradation proteins and a failure causes pathological conditions [17].

HSPs protect synaptic function in the nervous system from environmental insults or pathological factors [18–20] (reviewed in [21]), and are also associated to neurodegenerative diseases, aberrant protein-induced neurotoxicity and disease progression [13]. The sHSPs family is involved as a non-canonical role in *Drosophila* development and other biological processes such as synaptic transmission [22]. However, its implication in synaptic dynamics during development has not been described yet. Synapse number can be altered due to the influence of physiological parameters (aging, hormonal state, exercise) [23–26], pathological (neurodegenerative process) [27, 28] or induced conditions (mutants) [29] which alter cellular components and pathways [30]. The imbalance between the pro- and anti-synaptogenic pathways modulates the number of synapses [30]. The neuromuscular junction (NMJ) of *Drosophila melanogaster* is a stereotyped structure well established for the study of synapses [31]. Most of the molecules involved in synaptic transmission are conserved between *Drosophila* and vertebrates thus, this model system is well established for the study of synapses [32].

Here, we study the contribution of two *sHsps*, *sHp23* and *sHsp26* in the development of the CNS and synapse modulation. *sHsp23* and *sHsp26* are expressed in the CNS during the development [10, 33, 34] but their function remains unclear. In addition, we describe the function of CG1561, named Pinkman (Pkm), as a novel putative kinase that interacts with sHSP23 and sHSP26. Pkm regulates expression and protein stability and participates in the establishment of synapse number during development.

## Materials and methods

### *Drosophila* strains

Flies were maintained at 25°C in fly food in cycles of 12 hours of light and 12 hours of darkness. The following stocks were used: *UAS.LacZ* (gift from Dr. Wurz). Fly stocks from the Bloomington Stock Center: *Gal4.D42* (BL-8816), *UAS.Hsp23* (BL-30541), *P{UAS-mLex-A-VP16-NFAT}H2/TM3, Sb1* (BL- 66543), *P{LexAop-CD8-GFP-2A-CD8-GFP}2; P{UAS-mLex-A-VP16-NFAT}H2, P{lexAop-rCD2-GFP}3/TM6B, Tb1* (BL-66542). Fly Stocks from Vienna Stocks Center: *Hsp26-GFP-V5-Flag* (VDR318685), *UAS.Hsp26RNAi* and *UAS.CG1561RNAi* (VDR106503 KK (1), VDR32634 GD (2) and VDR32635 GD (3)). Fly Stocks from the FlyORF Zurich ORFeome Project: *UAS.Hsp26* (F000796).

### *Drosophila* dissection and immunostaining

*Drosophila* third instar larvae were dissected in phosphate-buffered saline (PBS) and fixed with PFA 4% in phosphate-buffered saline (PBS). Then, the samples were washed with PBST (PBS with 0.5% Triton X-100) and blocked with 5% bovine serum albumin (BSA) (Sigma) in PBST. We quantified the total number of active zones per NMJ of third instar larvae. We used the binary system Gal4/UAS (Brand & Perrimon, 1993) to drive all genetic manipulations to motor neurons (D42-Gal4). Actives zones were visualized using a mouse monoclonal antibody nc82 (1:20, DSHB, IA) which identifies the protein Bruchpilot, a presynaptic element. Neuronal membranes were visualized with rabbit anti-HRP (1:300, Jackson ImmunoResearch, West Grove, PA). Fluorescent secondary antibodies were Alexa 488 (goat anti-mouse, 1:500, Molecular Probes, Eugene, OR) and Alexa 568 (goat anti-rabbit, 1:500, Molecular Probes). Larvae were mounted in Vectashield medium (Vector Labs, Burlingame, CA). Synapse quantifications were obtained from the NMJ *Drosophila* model in muscle fiber 6/7 of the third abdominal segment only to regulate inter-individual data variability.

To localize sHSP23 or sHSP26, third instar larval brain or NMJ were dissected. We use an Hsp26-GFP-V5 fusion construct. sHSP23 was visualized using an anti-Hsp23 (Sigma-Aldrich S 0821) (1:500), and sHSP26 was visualized using anti-V5 (1:50) (Invitrogen 1718556) and anti-GFP mouse (1:50) (Invitrogen A11122). *Drosophila* brains were mounted in Vectashield with DAPI medium (Vector Labs, Burlingame, CA).

### Image acquisition

Confocal Images were acquired at 1024x256 resolution as serial optical sections every 1 µm. We used a 63x objective with a Leica Confocal Microscope TCS SP5 II (Mannheim, Germany). We used IMARIS software (Bitplane, Belfast, UK) to determine the number of mature active zones with the 'spot counter' module.

We visualized Hsp23-Hsp26 co-localization and CaLex signal in ventral ganglia cells of third instar larva brains. We acquired brain images at 1024x1024 resolution as serial optical sections every 1µm at 20x objective. We acquired ventral ganglia cells images at 1024x1024 resolution, 63x objective with magnification of 2.5. We processed the images and analyzed them with LAS-AF (Leica Application Suite software).

### Antibody generation

To detect sHsp26 protein in western blot we generated (Abmart) a mouse monoclonal antibody against the sHsp26 peptide sequence: GKENGAPNGKDK MSLSTLLSLVDELQEPRSP IYELGLGLHPHSRYVLPLGTQQRRSINGCPCASPICPSSPAGQVLALRREMANRNDIHWPAT AHVGKDGFQVCMDVAQFKPSELNVKVVDDSILVEGKHEERQDDHGHIMRHFVRRYKVPDGYK AEQVVSQLSSDGVLTVSIPKPQAVEDKSKERIIQIQQVGPAHLNVKANESEVK**GKENGA PNGKDK**

### Co-Immunoprecipitation

For biochemical assays, 5–10 adult fly heads were lysed in immunoprecipitation lysis buffer (NaCl 150 mM, 0,1% Tween-20 (Polyoxyethylene sorbitane monolaureate), TBS pH 7.5). We incubated Protein A/G agarose beads overnight at 4˚C with 2 µl of the indicated antibody or control IgG (1:100), followed by incubation at 4˚C for 1 h with supernatants. We washed the beads and resuspended in 1× SDS–PAGE loading buffer for western blot analysis in a 4%–12% gradient SDS-PAGE for the detection of sHSP23 and sHSP26. After electro-blotted onto nitrocellulose 0.45 µM (GE Healthcare) 100V for 1 hour, we blocked the membranes in

TBS-Tween-20 buffer with 5% BSA. We incubated the membranes overnight at 4˚C in constant agitation with anti-Hsp23 antibody (1:1000) (Sigma-Aldrich S0821), anti-Hsp26 (1:1000) (Abmart) We visualized the antibody-protein interaction by chemoluminescence using IRDye Secondary Antibodies anti-mouse (IRDye 800CW, LI-COR), anti-rabbit (IRDye 680 RD, LI-COR) and developed with Odyssey equipment. We used three RNAi tools to downregulate *pkm* expression to replicate this condition. *pkm* RNAi 2 was selected to do the rest of experiments due to the evidences we obtained in the blot assay.

## Western blot

5–10 head samples were treated with lysis buffer (TBS1x, 150mMNaCl, IP 50x) and then homogenized and centrifuged 13500 rpm for 5 minutes. After selecting the supernatant we added NuPage 4x (Invitrogen by Thermo Fisher Scientific) and ß-mercaptoethanol 5%. Western blot analysis samples were run in a 4%–12% gradient SDS-PAGE for the detection of sHSP23 and sHSP26. After electro-blotted onto nitrocellulose 0.45 μM (GE Healthcare) 100V for 1 hour, we blocked the membranes in TBS-Tween-20 buffer with 5% BSA. We incubated the membranes overnight at 4˚C in constant agitation with anti-Hsp23 antibody (1:1000) (Sigma-Aldrich S0821), anti-Hsp26 (1:1000) (Abmart) We visualized the antibody-protein interaction by chemo luminescence using IRDye Secondary Antibodies anti-mouse (IRDye 800CW, LI-COR), anti-rabbit (IRDye 680 RD, LI-COR) and developed with Odyssey equipment. Tubulin were used as a control.

## Gene expression analysis with qPCR

10–15 head tissue samples were treated and homogenized with Trizol (Ambiend for Life techonologies). Chloroform was added and then centrifuged 13000 rpm at 4˚C for 15 minutes. After discarding the supernatant, the RNA was treated with Isopropanol and then centrifuged 13000 rpm at 4˚C for 10 minutes and washed with 75% Ethanol. RNA pellet was dissolved in DNAase RNAase free water. Then we performed a transcriptase reaction and a qPCR assay using *Rp49* gene as a control. Primers for *sHsp23*, *sHsp26*, *Pinkman* and *Cat* were used: *sHsp23* Fv (5′-3′) TGCCCTTCTATGAGCCCTAC, *sHsp23* Rv (3′-5′) TCCTTTCCGATTTTCGACAC, *sHsp26* Fv (5′-3′) TAGCCATCGGGAACCTTGTA, *sHsp26* Rv (3′-5′) GTGGACGACTCCATCT TGGT, *pkm* Fv (5′-3′) TCGTGCTGGAGGATCTGTCTT, *pkm* Rv (3′-5′) CCCGGCCAATGATATA GCAT, *Catalase* Fv (5′-3′) TTCGATGTCACCAAGGTCTG, *Catalase* Rv (3′-5′) TGCTCCACCTCA GCAAAGTA, *rp49* Fv (5′-3′) CCATACAGGCCCAAGATCGT, *rp49* Rv (5′-3′) AACCGATGTTGGGCATCAGA.

## Statistics

To analyze the data, we used GraphPad Prism 6 GraphPad Software, La Jolla, CA). Data are shown as mean ± SD. Statistical significance was calculated using D´Agostino & Pearson normality test and a Student's two-tailed t-test with Welch-correction. In case data were not normal, we performed a Student´s two-tailed t-test with Mann–Whitney-U correction. For multiple comparisons, we used One-way ANOVA test with Bonferroni post-test. [*]p value ≤ .05; [**] p value ≤ .01; [***] p value ≤ .001; [****] p value<0,0001. p value > .05 were not considered significant.

## Technical considerations

Each experiment condition has its own control sample to reduce external variables.

## Results

### Heat shock proteins modify synapses in CNS

To determine the effect of HSPs in synaptogenesis, we used the UAS/Gal4 *Drosophila* binary expression system [35] to modify *Hsps* expression in motor neurons using *D42-Gal4* lines. We used UAS-RNAi lines knockdown *sHsp20*, *sHsp22*, *sHsp23*, *sHsp26*, *sHsp27*, *sHsp40*, *Hsp67 Ba*, *sHsp27 Bc*, *Hsp70 Aa*, *Hsp70 Ba* and *Hsp90* (**Fig 1A**). To visualize the number of active zones in the NMJs we used anti-bruchpilot (brp) antibody. The quantification of the active zones revealed that the knockdown of *sHsp20*, *sHsp22*, *sHsp26*, *sHsp27*, *sHsp40* and *Hsp90* during development provoked a reduction in synapse number. In addition, we tested the effect in synapse number of *sHsp23*, *sHsp26* and *Hsp70* overexpression (**Fig 1B**). The results show that the upregulation of *sHsp23*, *sHsp26 or Hsp70 de*crease the number of active zones (**Fig 1B**).

We focused on the role of two sHSPs, sHSP23 and sHSP26, due to their potential role as non-canonical-sHSPs in the CNS and their unexplored implication in synapses modulation. The upregulation of *sHsp23* in presynaptic neurons causes a reduction in synapse number (**Fig 1B**). In addition, *sHsp26* knockdown or upregulation induces a reduction in synapse number (**Fig 1A** **and Fig 1B**). Thus, the results suggest that sHSP23 is not required for synapse formation but in excess it is detrimental for the neuron and causes a reduction of synapse number during development. Besides, modification in any direction of *sHsp26* expression affects to the correct establishment of synapse number during development, suggesting that sHSP26 fine control is required during development for synapse organization.

According to the interactome (flybase) both chaperones are predicted to physically interact with each other [36] (**Fig 1C** **and Fig 1D**). Furthermore, sHSP23 and sHSP26, both interact with: CG11534, CG43755 and Pkm (CG1561) proteins [36] (**Fig 1C** **and Fig 1D**).

### sHSP23 and sHSP26 colocalize in neurons and interact physically

To determine the expression and subcellular localization of sHSP23 and sHSP26 proteins in larval brain we used a green fluorescent reporter tagged form of sHSP26 (HSP26-GFP-V5) and we generated a monoclonal specific antibody against sHSP26 (**S1 Fig**). We dissected third instar larvae brain and visualized both sHSPs. The data show that sHSP23 and sHSP26 localize in the cytoplasm of CNS cells, in particular in the optic lobes and the central nerve cord (**Fig 2A and 2B'**). The co-localization of both proteins occurs in neuroblasts and also in ganglion mother cells and differentiated neurons, compatible with a general role in nervous system development.

To further analyze the presence and co-localization of sHSP23 and sHSP26 we analyzed larvae NMJs (**Fig 2C–2F'**). The confocal images show an accumulation and colocalization of sHSP23 and sHSP26 throughout the NMJ but particularly intense in the synaptic buttons (**Fig 2C–2F'**). This observation is compatible with a role in synaptic activity as most of the active zones are in the synaptic buttons.

### sHSP23 and sHSP26 interact physically

In general, sHSPs proteins exhibit regions susceptible of posttranslational modifications (PTMs) which favor their oligomerization and alter the affinity of interaction by co-chaperones [17, 37]. Since, this mechanism maintains the activity of sHSPs it has been proposed that it regulates their function [17].

The results show that both proteins are localized in the same sub cellular compartments. To determine if both chaperones interact physically, we performed a co-immunoprecipitation assay. We used head protein extracts that were incubated with specific antibodies to

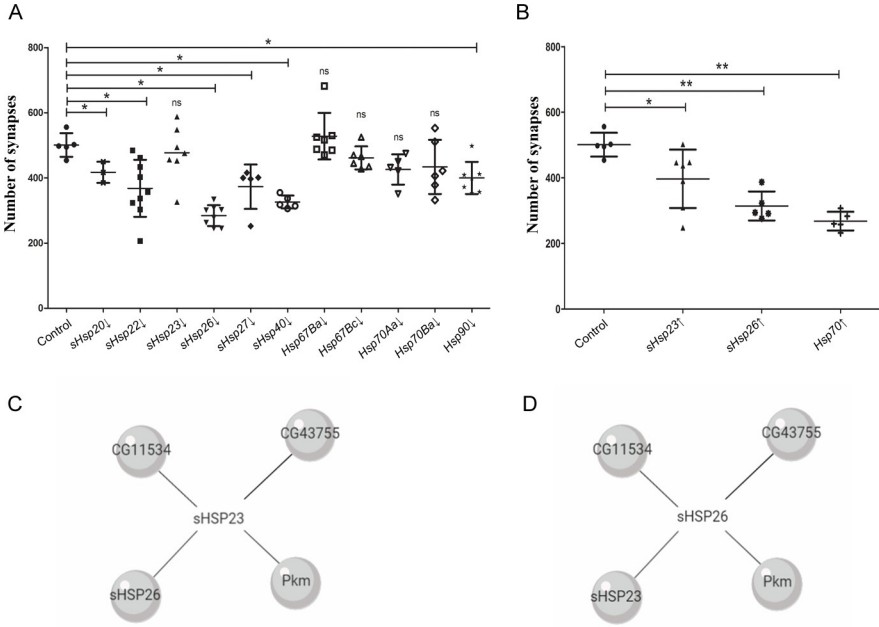

**Fig 1. Small heat shock proteins modulate synapses during *Drosophila* development.** Synapses quantification screening with *sHsps* genetic tools under *D42* driver expression. **(A)** Synapses modulation were detected by *sHsp20* RNAi (*sHsp20↓*), *sHsp22* RNAi (*sHsp22↓*), *sHsp23* (*sHsp23↓*), *sHsp26* RNAi (*Hsp26↓*), *sHsp27* RNAi (*Hsp27↓*), *Hsp40* RNAi (*Hsp40↓*), *Hsp90* RNAi (*Hsp90↓*), **(B)** *UAS.sHsp23* (*Hsp23↑*) *UAS.sHsp26* (*Hsp26↑*) and *UAS.Hsp70* (*Hsp70↑*) samples. One-way ANOVA test with Dunn's multiple comparisons post-test. *p value ≤ .05; ** p value ≤ .01; *** p value ≤ .001. p value > .05 were not considered significant. Error bars show S.D. **(C)** Diagram of *sHsp23* interactome and **(D)** diagram of *sHsp26* interactome form http://flybi.hms.harvard.edu/results.php.

specifically immobilize each sHSP in Protein A/G agarose beads. Samples were pre-cleared with untagged beads to avoid unspecified binding. Agarose beads were incubated with HSP23 or HSP26 antibody overnight. Head extract proteins and antibody-bound beads were incubated 1 hour. We revealed the western blot membranes with sHSPs antibodies and the results show that sHSP23 (**Fig 2G** lane 2) immunoprecipitation also precipitates HSP26, and vice versa (**Fig 2G** lane 3). Both specific bands are corroborated in the input positive control (**Fig 2G** lane 1) and the lack of signal in the negative control (**Fig 2G** lane 4) 22c10 antibody). These results confirm the physical interaction between sHSP23 and sHSP26 (**Fig 2H**), and it is consistent with their co-localization in the motor neuron buttons.

## Pkm interacts with sHSP23 and sHSP26 and modulate synapse number

CG11534, CG43755 and Pkm proteins have been postulated that interact with both sHSP23 and sHSP26 (Flybase). They have unknown functions but predicted to have protein kinase like activity (Flybase, http://flybi.hms.harvard.edu/results.php). HSPs posttranslational modifications modulate their function [17] and therefore, we quantified the number of active zones in the NMJ upon knockdown of each candidate gene. The knockdown of *pkm* in motor neurons increases synapses number while we could not find any significant change for *CG11534* and *CG43755* knockdown (**Fig 3A**). In consequence, we focused our study in *pkm* as a candidate gene to interact with *sHsp23* and *sHsp26* in nervous system development.

*pkm* is a novel gene that encodes a protein with kinase like domains (Flybase) (**Fig 3B**). Pkm is reported to physically interact with sHSP23 and sHSP26 [36] (**Fig 3C**). Posttranslational changes modulate chaperones and co-chaperones interaction and activity [17], thus it is

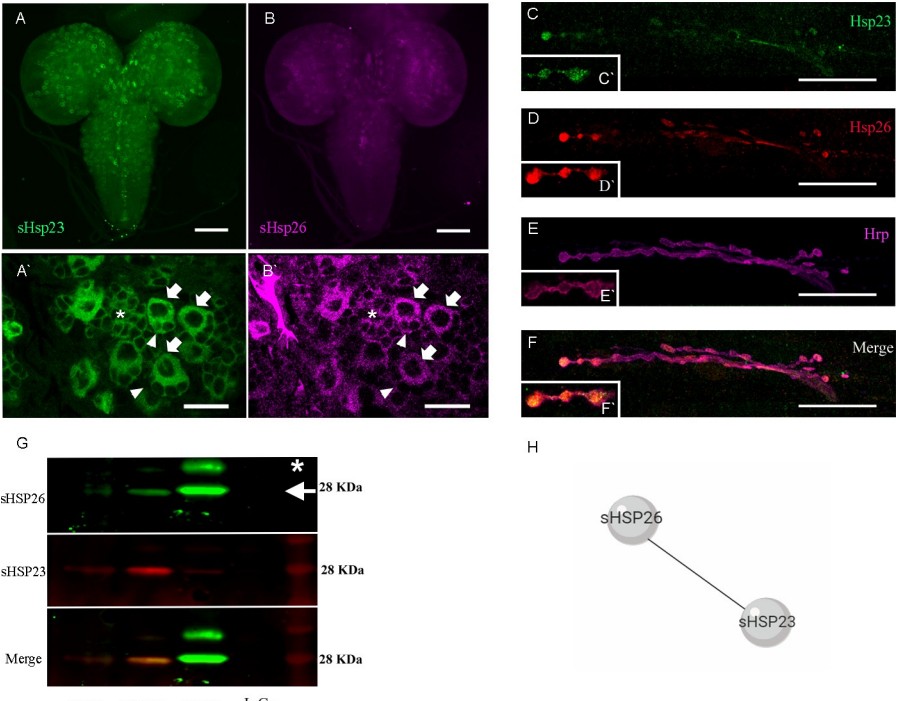

**Fig 2. sHSP23 and sHSP26 colocalize in CNS. (A-F)** Confocal microscopy images of 3ʳᵈ instar *Drosophila* larval brain and NMJs. **(A)** sHSP23 is labeled with anti-GFP antibody driven by *D42-Gal4* to visualize its expression in brain regions (magenta). Scale bar size 100 um. **(B)** sHSP26 is stained with anti-sHSP23 (green). **(A'-B')** Magnification images of larval brain. Arrows indicate neuroblast, arrowheads indicate ganglion mother cells and asterisk indicate neurons where sHSP23 and sHSP26 colocalize in the cytoplasm. Scale bar size 100 um. **(C-F)** sHSP26 is labeled with anti-GFP antibody driven by *D42-Gal4* to visualize its expression in NMJ (red), sHSP23 is stained with anti-sHSP23 (green) and neuronal membrane is detected with anti-HRP staining (magenta). Scale bar size 50 um **(C'-F')** Magnification images of synaptic boutons in NMJ. **(G)** Co-Immunoprecipitation assay membrane revealed with sHSP26 (green, arrow) and sHSP23 (red) antibodies in control samples. Fly heads were lysed in immunoprecipitation lysis buffer and incubated with protein A/G agarose beads previously treated with sHSP23 or sHSP26 antibody and IgG antibody as a control. The samples were prepared for western blot analysis. The antibody-protein interaction is visualized by chemoluminescence. Molecular weights are indicated in all the membrane images. * Unknown/ unspecific band **(H)** sHSP23 and sHSP26 interaction diagram.

suggested that these mechanisms represent a system to modulate chaperone dynamics. Accordingly, we did immunoprecipitation assays to determine if Pkm was necessary for the sHSP23 and sHSP26 complex formation (**Fig 3D**). The results revealed that *pkm* knockdown does not modify the interaction between chaperones, thus *pkm* expression is dispensable for sHSP23-sHSP26 physical interaction.

Furthermore, we tested if the expression of *pkm* could modulate the expression of *sHsps*. To evaluate transcription, we did quantitative PCR (qPCR) experiments of *pkm*, *sHsp23*, *sHsp26* and *catalase (Cat)* as a positive control. The results show that *pkm* RNAi proved effective since its transcription is drastically reduced (Fig 4A). On the other hand, s*Hsp23*, but not *sHsp26*, expression is largely increased (**Fig 4A**). In order to confirm the transcriptional results, we analyzed the total protein amount of sHSPs upon *pkm* knockdown and those of sHSP23 and sHSP26 by Western blot assays (**Fig 4B**). To silence the expression of *pkm* we used three RNAi tools to replicate this condition and to reaffirm the regulation changes that we found with the qPCR. The knockdown of *pkm* with *pkm* RNAi 2 tool provokes an increase in sHSP23 and sHSP26 proteins (**Fig 4C**). The data for *sHsp23* are consistent with the qPCR assays and suggest that *pkm* is necessary to restrict *sHsp23* expression, while *sHsp26* expression is

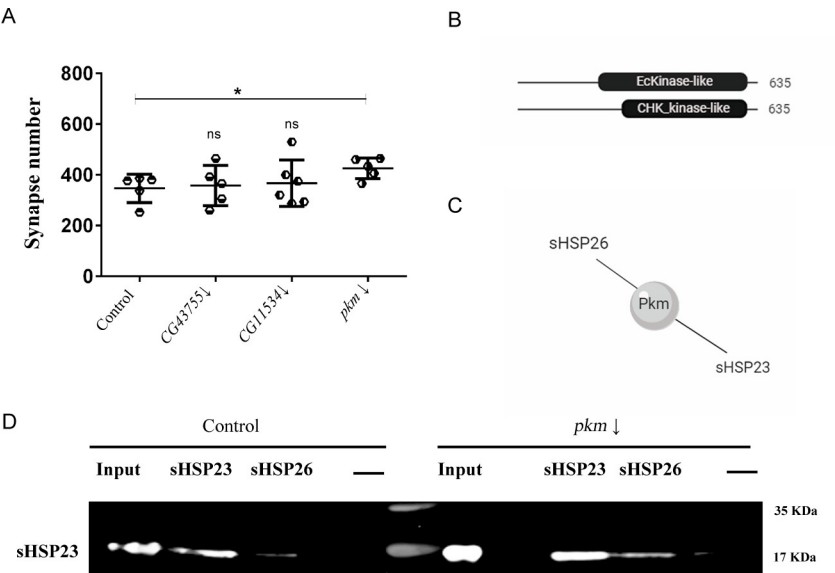

**Fig 3. Pkm does not affect to sHSP23-sHSP26 interaction. (A)** Quantification of synapse active zones in the NMJ is shown for the knockdown of all candidate genes genotypes: *CG43755* RNAi (*CG43755↓*), *CG11534* RNAi (*CG11534↓*) and *pkm* RNAi (*pkm↓*). One-way ANOVA test with Bonferroni post-test* $P < 0.05$. Error bars show S.D. **(B)** *pkm* contains a EcKinase like (Ecdysteroid kinase-like) domain between 257–545 aa sequence and a CHK_kinase like (Choline kinase-like) domain between 346–543 aa sequence. **(C)** Diagram of Pkm interactome. Pkm physically interacts with sHSP23 and sHSP26. **(D)** Co-immunoprecipitation assay membrane revealed with sHSP23 antibody in control and *pkm* RNAi samples. Molecular weights are indicated.

independent of *pkm* expression but protein accumulations is increased upon *pkm* knockdown. These data are compatible with sHSP26 posttranslational modifications mediated by Pkm to control protein stability and degradation.

## Pkm modulates synapse number

Small chaperones work as dimers or oligomers to modulate their activity [38], therefore sHSPs protein-protein interaction opens a potential activity as a complex. Since sHSPs family is characterized by forming oligomer assemblies based on dimers joined [39], we investigated the coordinated effect of sHSPs and Pkm.

*pkm RNAi* causes an increase of synapse number in development (**Figs 3A, 4D** and **4E**), moreover *sHsp23* upregulation reduces synapse number and knockdown does not change synapse number during development (**Fig 1**). We combined *pkm RNAi* and *sHsp23 upregulation* in neurons and we observed an increase in synapse number sample comparable to *pkm RNAi* alone, suggesting that the effect of *pkm* RNAi for synaptogenesis during development is mediated by sHSPs (**Fig 4D**). These results suggest that the synaptogenic effect of *pkm* knockdown is not restricted to *sHsp23* upregulation (**Fig 4A–4C**), thus we investigated the contribution of *sHsp26* in combination with *pkm*. Protein quantification experiments show that sHSP26 is accumulated upon *pkm RNAi* expression but in a lesser extent than sHSP23. To demonstrate that sHSP26 is the limiting factor in sHSP23/26 complex we upregulated *sHsp26* together with *pkm RNAi*. Synapse quantifications show that upregulation of *sHsp26* can further increase synapse number in *pkm RNAi* background (**Fig 4E**). These results suggest that sHSP26 is a limiting factor for synaptogenesis in development and *pkm* reduction contributes to stabilize sHSP26 partially but *sHsp26* upregulation causes complementary synaptogenesis.

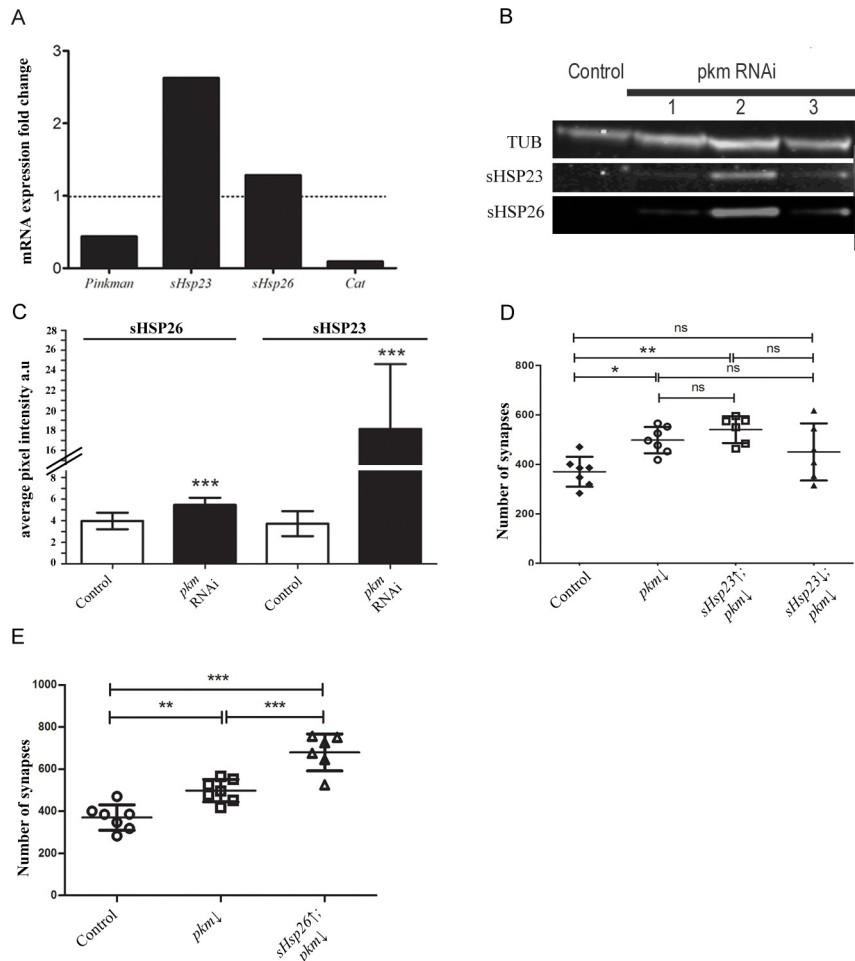

**Fig 4. sHSPs amount is regulated by the novel candidate gene *pkm*. (A)** qPCR assay of *pkm* RNAi sample measuring mRNA expression fold change of 3$^{rd}$ instar larval of *pkm*, *sHsp*23, *sHsp*26 expression and *Cat* as positive control, normalized with *Rp49* as a control. **(B)** Western blot assay of control and *pkm* RNAi (*pkm↓*) samples stained against sHSP23 and sHSP26. We used three RNAi tools to confirm the protein amount changes under *pkm* downregulation condition. *pkm* RNAi 2 was selected due to its efficacy. Tubulin was used as a control. **(C)** Mean Intensity sHSP23 and sHSP26 signal are shown for control and *pkm* RNAi (*pkm↓*) samples. Unpaired T-test Welch´s correction* P<0.05. Error bars show S.D. **(D)** Quantification of synapse active zones in the NMJ is shown for the combination of *sHsp23* and *pkm*, *sHsp26* downregulation under *D42* driver expression. **(E)** Synapse number quantification in NMJs after *sHsp26* upregulation and pkm downregulation. Unpaired T-test Mann Whitney post-test * p value<0.05; p value > .05 were not considered significant. Error bars show S.D.

To further determine if sHSPs protein interaction and *pkm* modulation contribute to synaptogenesis, we altered *sHsps* expression together and counted synapse number in NMJs. The joint upregulation of *sHsp23* and *26* induces an increase in synapse number (**Fig 5A**). This increase contrasts with the reduction elicited by each *sHsp* when upregulated separately (compare to Fig 1A). Moreover, the upregulation of both chaperones in combination with *pkm* knockdown maintains the drastic increase in synapse number and, actually, is of a larger magnitude than the *pkm* knockdown by itself (**Fig 5A**). This observation is compatible with a functional interaction of Pkm kinase with at least one of the two sHSP analyzed here.

Finally, the combined knockdown of both *sHsps* does not alter synapse number. This result suggests that only the modulation of one single chaperone of this combo produces an imbalance that triggers synapse loss. If both chaperones are reduced in combination, there is no

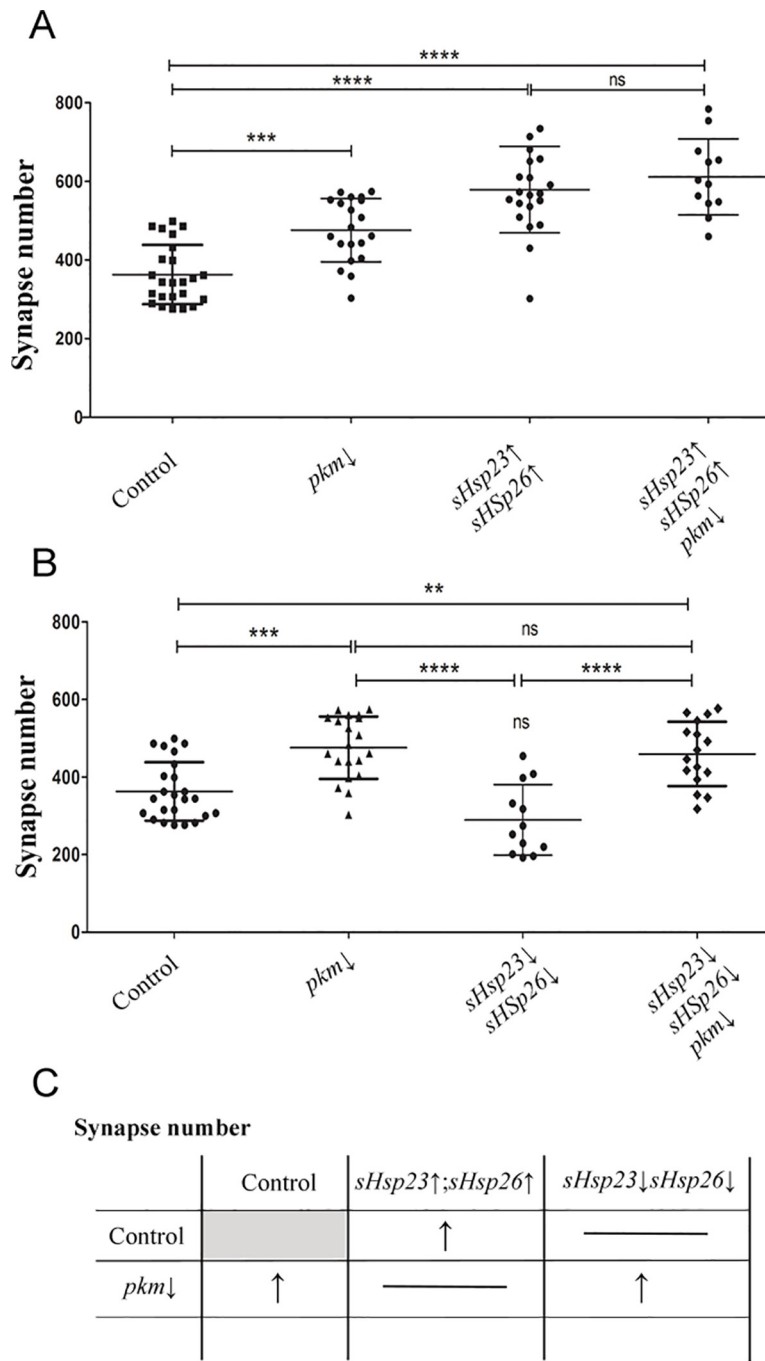

**Fig 5. Pkm activity is restricted by *sHsps* expression.** Quantification of synapse active zones in the NMJ is shown for the combination of *sHsps* expression and *pkm* downregulation under *D42* driver expression: **(A)** *pkm* RNAi (*pkm↓*), UAS.sHsp23; UAS.sHsps26 (*sHsp23↑*; *sHsp26↑*), UAS.sHsp23; UAS.sHsps26/pkm RNAi (*sHsp23↑*; *sHsp26↑/pkm↓*), **(B)** UAS.sHsp23 RNAi; UAS.sHsps26 RNAi (*sHsp23↓*; *sHsp26↓*), UAS.sHsp23 RNAi; UAS.sHsps26 RNAi /pkm RNAi (*sHsp23↓*; *sHsp26↓/pkm↓*). One-way ANOVA test with Bonferroni post-test. *p value ≤ .05; ** p value ≤ .01; *** p value ≤ .001. p value > .05 were not considered significant. Error bars show S.D. **(C)** Summary table for the combination of *sHsps* expression and *pkm* downregulation.

effect what supports the idea that the equilibrium between sHSP23 and sHSP26 is relevant. Moreover, *sHsp23*, *sHsp26* and *pkm RNAi co*-expression show an increase in synapse number

(**Fig 5B**). Thus, we conclude that the effect of *sHsps* upregulation surpasses *Pkm* contribution but, *sHsps* knockdown is not sufficient to prevent the increase in synapse number caused by *pkm* knockdown (**Fig 5B**).

As a result, we suggest that sHSP23 and sHSP26 together form a complex that promotes synapse formation in presynaptic neurons, Pkm is an anti-synaptogenic element in neurons through, but not restricted to, the modulation of *sHsp23* and *sHsp26* (**Fig 5C**).

## Neuronal activity correlates with synapses changes caused by sHSPs and Pkm

Changes in synapse number are expected to reflect on neuronal activity. To evaluate the cellular effect of the observed synapse number changes, we took advantage of CalexA system (Calcium-dependent nuclear import of LexA) to perform a functional assay in motor neurons [40].

CalexA is a tracing system to label neuronal activity based on calcium/NFAT signaling and the two binary expression systems *UAS/Gal4* and *LexA/LexAop* [41]. We used specific lines to drive a modified NFAT form to motor neurons (*D42.Gal4* and *P{LexAop-CD8-GFP-2A-CD8-GFP}2; P{UAS-mLexA-VP16-NFAT}H2, P{lexAop-rCD2-GFP}3/TM6B, Tb1*). The accumulation of $Ca^{2+}$ due to the action potentials activates calcineurin which dephosphorylates NFAT, provoking its import into the nucleus. NFAT binds to LexAop sequence and induces the expression of a *GFP* reporter gene (**Fig 6A**). Therefore, *GFP* signal becomes a reporter of neuronal activity.

To evaluate neuronal activity and *sHsps* and *pkm* expression we measured the signal of the *GFP* reporter in larva brains (**Fig 6B–6E**). The *sHsp23* and *sHsp26* upregulation increases *GFP* signal in cells of ventral nerve cord (**Fig 6C and 6E**) which correlates with an increase in the number of synapses (**Fig 5A**). By contrast, *pkm* knockdown reduces CalexA reporter signal in motor neurons (**Fig 6D and 6E**) which correlates with its anti-synaptogenic role in motor neurons. The results indicate that Pkm contribution is not limited to sHSP23 and sHSP26. Here *pkm* knockdown reproduces the effect of small chaperones upregulation on neuronal activity. Besides, we have shown that *pkm RNAi* synaptogenic effect is not prevented by *sHsp23* and *sHsp26 RNAi* (**Fig 5B**). Therefore, additional Pkm targets participate in synapse formation and neural activity. Taking all these data together, we conclude that *sHsps* and *pkm* expression participate in synapse formation during development and neuronal activity.

## Discussion

Synapse regulation is a central event during nervous system development and adult life. Disruptions in the establishment of synapses is associated with morphological, cognitive and psychiatric disorders, but the precise mechanisms underlying these disorders remain unknown [42]. Changes in synapse structure and function are related to paralysis and muscular atrophy in amyotrophic lateral sclerosis (ALS) [43, 44], impairment of the neuromuscular junction function and therefore, motor decline [45] or social and cognitive behaviors related to autism [46]. Thus the study of relevant mechanisms for synapse formation during development is a need.

Chaperones participate in protein folding maintenance as a mechanism to regulate function and pathological conditions, but the specific contribution to synapse number during development was not addressed. Here we describe the combined contribution of two sHSPs (sHSP23 and sHSP26) to synapse formation and the modulation by a novel putative kinase protein Pkm. *sHsp23* mRNA and total protein amount increases upon *pkm* knockdown, suggesting a transcriptional negative regulation by Pkm.

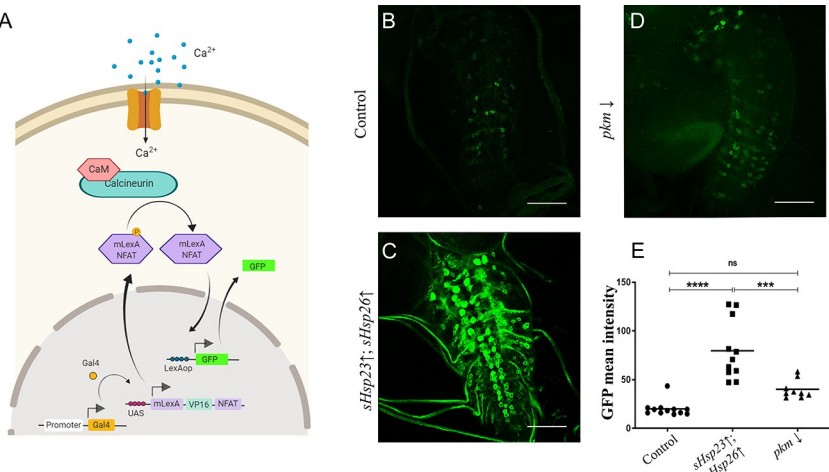

**Fig 6. sHSPs contribute to neuronal activity *GFP* signal. (A)** CalexA system labels neuronal activity based on calcium/NFAT signaling after a neuronal action potential and the two binary expression systems *UAS/Gal4* and *LexA/LexAop*. The accumulation of calcium activates calcineurin that dephosphorylates NFAT that are imported to the nucleus. NFAT binds to LexAop sequence and induces the expression of *GFP* reporter gene that correlates with neuronal activity. **(B-D)** Confocal microscopy images of 3$^{rd}$ instar *Drosophila* larval ventral nerve cord of **(B)** control, **(C)** *UAS.sHsp23; UAS.sHsps26* (*sHsp*23↑; *sHsp*26↑), **(D)** *pkm* RNAi (*pkm*↓) samples. **(E)** *GFP* mean intensity signal quantification is shown for *sHsps* expression and *pkm* downregulation under *D42* driver expression: *UAS.sHsp23; UAS.sHsps26* (*sHsp*23↑; *sHsp26*↑), *pkm* RNAi (*pkm*↓), One-way ANOVA test with Bonferroni post-test.; *** p value = 0.002. **** p value<0,0001;. p value > .05 were not considered significant. Error bars show S.D.

The results show that *sHsp26* mRNA levels do not show significant changes after *pkm* expression interference. However, *HSP26* in yeast is degraded via a ubiquitin/proteasome-dependent mechanisms [47], the results from western blot experiments show HSP26 protein accumulation upon *pkm RNAi* expression. Therefore, as Pkm is proposed to be a kinase, we cannot discard that Pkm could promote HSP26 post-transcriptional modifications (phosphorylation) to promote its degradation. According to the putative domains present in Pkm protein, there are no DNA binding domains and hence it is unlikely that Pkm acts as a transcription factor. We postulate that the transcriptional regulation of *sHsp23* is determined by transcription factors sensible to posttranslational modifications as direct targets of Pkm. However, the precise mechanisms and molecular details of Pkm-sHSPs relation require be further investigated.

Genetic modifications of one single *sHsp* (*sHsp23* or *sHsp26*) cause an imbalance in the equilibrium between both genes, as a consequence it causes a reduction in the number of synapses. These results suggest that single alterations in sHSPs are detrimental for the number of synapses. However, we propose that sHSP23 function in synaptogenesis requires forming a complex with sHSP26. *sHsp26* upregulation and downregulation modify synapse number in the same direction (synapse number reduction). However, sHSP23 is the one modulated by Pkm but *sHsp23* downregulation does not change synapse number. It is tempting to speculate that a reduction in sHSP23 does not affect to CNS development and it has a protective function more than synaptogenic. However, when sHSP23 and sHSP26 play together they modulate synapses, upregulation of both genes cause an increase in synapse number. Moreover, *pkm* knockdown increases synapse number but does not further increase synapse number upon *sHsps* overexpression, indicating that *pkm* effect on synapses is mediated by *sHsps* regulation. In addition, the combined silencing of *sHsps* does not alter synapse number, in line with the proposal of sHSPs equilibrium. But *pkm* knockdown increases significantly the number of synapses in *sHsps* knockdown background. These results suggest that *pkm* is a repressor of *sHsps*

and *pkm RNAi* counteracts the reduction of sHSPs. Thus we speculate with the hypothesis of sHSP26 acting as a synapse modulator and sHSP23 as a protective partner regulated by Pkm.

A direct consequence of *sHsps* modulation is a reduction in neuronal intracellular calcium levels in the brain, an indicator of neuronal activity. Co-overexpression of both sHSPs results in enhanced intracellular calcium activity directly associated to neuronal activity. Therefore, small chaperones are required for the formation of the correct synapse number in NMJs and also can stimulate brain activity. These results connect neural activity and chaperones, which are proteins that sense environmental changes and in consequence, link neural activity and environment during development. In particular, these two chaperons are associated to temperature changes [48, 49], environmental-stress-induced degeneration [50, 51] and lifespan [52]. Besides, maternal loading of sHSP23 determines embryonic thermal tolerance pointing to a physiological role during development [53]. All these evidences support that *sHsp* disruption during embryogenesis and development can be associated to physiological defects in adulthood, therefore Pkm-sHSPs contribution during development is proposed as a central mechanism for nervous system correct formation, function and response to environmental stress.

## Supporting information

**S1 Fig. Tools validation. (A-B)** To validate if the antibody that we generated against sHsp26 is specific, we knocked down *sHsp26* in the posterior compartment of wing imaginal disc (*engrailed-Gal4*) and visualized the specific domain with the co-expression of GFP. **(C)** The quantifications of pixel intensity show that anti-sHsp26 recognizes the reduction of *sHsp26* expression caused by *UAS-sHsp26 RNAi*. Unpaired T-test Welch´s correction **** p value<0,0001. Error bars show S.D.
(TIF)

**S1 Raw images.**
(PDF)

## Acknowledgments

We thank Professor Alberto Ferrús, Dr. F.A. Martín, María Losada, Patricia Jarabo and anonymous reviewers for critiques of the manuscript and for helpful discussions. Clemencia Cuadrado for fly stocks maintenance. We are grateful to the Vienna *Drosophila* Resource Centre, the Bloomington *Drosophila* stock Centre and the Developmental Studies Hybridoma Bank for supplying fly stocks and antibodies, and FlyBase for its wealth of information. We acknowledge the support of the Confocal Microscopy unit and Molecular Biology unit at the Cajal Institute for their help with this project. We acknowledge support of the publication fee by the CSIC Open Access Publication Support Initiative through its Unit of Information Resources for Research (URICI).

## Author Contributions

**Conceptualization:** Elena Santana, Sergio Casas-Tintó.

**Formal analysis:** Elena Santana, Teresa de los Reyes, Sergio Casas-Tintó.

**Funding acquisition:** Sergio Casas-Tintó.

**Investigation:** Elena Santana, Teresa de los Reyes, Sergio Casas-Tintó.

**Methodology:** Elena Santana, Teresa de los Reyes, Sergio Casas-Tintó.

**Project administration:** Sergio Casas-Tintó.

**Supervision:** Sergio Casas-Tintó.

**Writing – original draft:** Elena Santana, Teresa de los Reyes, Sergio Casas-Tintó.

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
