## [Decision Letter · Decision Letter 0]

19 Mar 2020

PONE-D-20-03396

Small heat shock proteins determine synapse number and neuronal activity during development

PLOS ONE

Dear Casas-Tinto,

First of all, my apologies for the delay in reply, due to 1) the fact that I could not get a 2nd reviewer for you MS and 2)  the corona crisis, which may have been cause of the  latter, but certainly also was the reason for my slow reply as I had to coordinate the research shutdown at our institute.

After careful consideration, in this case of 1 reviewer and myself going over the MS, we feel that it has merit but does not meet PLOS ONE’s publication criteria as it currently stands.

A main problem is that the paper is confusingly arranged (mis arrangements of figures etc), which made it impossible to me -as Editor- to evaluate the MS adequately. This was also noted by the reviewer, who nevertheless did think the content was potentially interesting, but who raised a number of major concerns.

Given the delay and the absence of a report form a 2nd reviewer, I therefore decided to give you the "benefit of the doubt" and to invite you to submit a revised version of the manuscript.

This MS should not only be better organised but also address ALL the questions raised by the reviewer.

I will check this personally before asking this reviewer to evaluate your revision.

Please note that this, by no means, does not imply that a Revision is will be automatically accepted.

We would appreciate receiving your revised manuscript by May 03 2020 11:59PM. To enhance the reproducibility of your results, we recommend that if applicable you deposit your laboratory protocols in protocols.io, where a protocol can be assigned its own identifier (DOI) such that it can be cited independently in the future. For instructions see: http://journals.plos.org/plosone/s/submission-guidelines#loc-laboratory-protocols

We look forward to receiving your revised manuscript.

Kind regards,

Harm H Kampinga

Academic Editor

PLOS ONE

Journal Requirements:

3. Thank you for stating the following financial disclosure:"The funders had no role in study design, data collection and analysis, decision to publish, or preparation of the manuscript."

Please provide an amended Funding Statement that declares *all* the funding or sources of support received during this specific study (whether external or internal to your organization) as detailed online in our guide for authors at http://journals.plos.org/plosone/s/submit-now.  

Reviewers' comments:

Reviewer's Responses to Questions

**Comments to the Author**

1. Is the manuscript technically sound, and do the data support the conclusions?

Reviewer #1: Partly

2. Has the statistical analysis been performed appropriately and rigorously? 

Reviewer #1: Yes

3. Have the authors made all data underlying the findings in their manuscript fully available?

Reviewer #1: Yes

4. Is the manuscript presented in an intelligible fashion and written in standard English?

Reviewer #1: Yes

5. Review Comments to the Author

Reviewer #1: In their manuscript Reyes et al. describe the role of two Small Heat Shock Proteins, namely sHSP23 and sHSP26, in synaptogenesis and neuronal activity. The authors show that when overexpressed alone both sHSP23 and sHSP26 lead to a decrease in synapse number; by contrast, when expressed together, sHSP23 and sHSP26 increase the synapse number. These data suggest that imbalances in the expression of these two sHSP may be detrimental, while their tightly regulated expression is required for synaptic development. The article goes further in describing that a novel putative kinase protein, named by the authors Pinkman (Pkm) is responsible for the modulation of the complex sHSP23-sHSP26 by transcriptionally repressing sHSP23.

The article is interesting, making use of well-designed experiments to demonstrate the interactions and role of the proteins described. However, the numerous errors in the presentation of the manuscript (see minor comments) has rendered it difficult to revise and show a lack of professionalism. Moreover, there are several points that need to be addressed prior to manuscript publication.

Major comments:

1- The manuscript contains several typos and grammatical errors. Please revise thoroughly.

2- In Fig. 2G, it is unclear how the immunoprecipitation was performed. Why it is represented twice in the same gels? Is it normal that sHSP26 gives 2 bands? The figure should be better described both in the text as in the legend.

3- In Fig. 3C, the authors show an interactome diagram of Pkm with both sHSPs while in the text they state that Pkm immunoprecipitates with both sHSPs. If the authors demonstrated the immunoprecipitation, they must show the blot/results and not just an interactome diagram. If this was demonstrated by another group, they should refer to the original study. In any case experimental demonstration of this interaction should be shown.

4- In Fig. 4A, the authors show the RNA expression of several genes upon Pkm silencing. Figure and legend are not accurate. What is shown is the fold-change compared to control and this should be clearly described in the figure and figure legend.

5- In Fig. 6B-E, the authors look at the functional effects of sHSP overexpression or Pkm silencing. They state that overexpression of sHSP23 and sHSP26 lead to an increase in CalexA reporter signalling, which is easily observed in Fig. 6C and E. On the other hand, they state that Pkm silencing leads to a reduced CalexA signalling; however, when looking at Fig. 6D and E, we see a slight increase in the GFP signal (Fig. 6D) and the quantification is similar to control levels (Fig. 6E). The authors should verify this result and interpret accordingly.

Minor comments:

1. The figures are erroneously numbered: Fig.1 is labelled as Fig. 5; Fig. 2 as Fig.6; Fig. 3 as Fig.1; Fig. 4 as Fig.2; Fig. 5 as Fig.3; Fig. 6 as Fig.4. This is unpleasant!

2. In Fig. 1C and D, one of the proteins represented in the interactome diagrams is labelled CG143755, while in the text the authors refer to it as CG43755. Furthermore, the authors refer to a protein named CG1561, while in these diagrams it is labelled as Pmk. The authors should already state the new name of this protein in the text or change the labels in the diagrams.

3. In Fig. 2A and B, the figure labelling doesn’t match the figure legend. In A, we see the label for sHSP23 and in B for sHSP26 while in the legend, the authors state the contrary.

4. In Fig. 4C-E, there is difference between the figure labelling and the legend description. Fig. 4C shows the quantification of the blot for both sHSP23 and sHSP26 while in the legend they are separated as C and D. Consequently, Fig. 4D is described as 4E in the legend, and so on. Furthermore, the summary table illustrated in Fig. 4D doesn’t bring much information and could be removed.

6. PLOS authors have the option to publish the peer review history of their article (what does this mean?). If published, this will include your full peer review and any attached files.

Reviewer #1: No

---

## [Author Response · Author response to Decision Letter 0]

6 Apr 2020

Review Comments to the Author

Reviewer #1: In their manuscript Reyes et al. describe the role of two Small Heat Shock Proteins, namely sHSP23 and sHSP26, in synaptogenesis and neuronal activity. The authors show that when overexpressed alone both sHSP23 and sHSP26 lead to a decrease in synapse number; by contrast, when expressed together, sHSP23 and sHSP26 increase the synapse number. These data suggest that imbalances in the expression of these two sHSP may be detrimental, while their tightly regulated expression is required for synaptic development. The article goes further in describing that a novel putative kinase protein, named by the authors Pinkman (Pkm) is responsible for the modulation of the complex sHSP23-sHSP26 by transcriptionally repressing sHSP23.

The article is interesting, making use of well-designed experiments to demonstrate the interactions and role of the proteins described. However, the numerous errors in the presentation of the manuscript (see minor comments) has rendered it difficult to revise and show a lack of professionalism. Moreover, there are several points that need to be addressed prior to manuscript publication.

We would like to apologize for these errors, we have amended all of them in this novel submission.

Major comments:

1- The manuscript contains several typos and grammatical errors. Please revise thoroughly.

We have revised the text and corrected typos and grammatical errors.

2- In Fig. 2G, it is unclear how the immunoprecipitation was performed. Why it is represented twice in the same gels? Is it normal that sHSP26 gives 2 bands? The figure should be better described both in the text as in the legend.

We have included further details about the immunoprecipitation protocol and design. We have deleted one of the gels as it was duplicated information.

Regarding the two bands for HSP26, we have first validated that the peptide used for the generation of the antibody is not contained in any other Drosophila protein. We have performed novel experiments to validate the antibody (S1 Fig) to confirm that anti-Hsp26 recognizes the reduction of hsp26 expression in RNAi experiments.

We have discarded the possibility of two hsp26 isoforms of different molecular weight, thus, we favour the hypothesis of post translational modifications that alter the migration of HSP26 in western blot, or an aggregation process resistant to the standard western blot conditions. Both possibilities require a great amount of work and will be solved in the future as these questions are out of the scope of this manuscript. 

3- In Fig. 3C, the authors show an interactome diagram of Pkm with both sHSPs while in the text they state that Pkm immunoprecipitates with both sHSPs. If the authors demonstrated the immunoprecipitation, they must show the blot/results and not just an interactome diagram. If this was demonstrated by another group, they should refer to the original study. In any case experimental demonstration of this interaction should be shown.

We have included the original references that show physical interaction between these proteins (Guruharsha KG et al. A protein complex network of Drosophila melanogaster. Cell. 2011;147(3):690-703). In addition, we tried to get a specific antibody against Pkm but we have not found any in the market. Thus, we generated a monoclonal antibody against Pkm (Abmart) and we have used it to detect Pkm in western blot/immunoprecipitation, but we did not get any positive result. Unfortunately, the antibody against Pkm does not work in western blot in our hands. 

4- In Fig. 4A, the authors show the RNA expression of several genes upon Pkm silencing. Figure and legend are not accurate. What is shown is the fold-change compared to control and this should be clearly described in the figure and figure legend.

We have corrected this in the text and figure

5- In Fig. 6B-E, the authors look at the functional effects of sHSP overexpression or Pkm silencing. They state that overexpression of sHSP23 and sHSP26 lead to an increase in CalexA reporter signalling, which is easily observed in Fig. 6C and E. On the other hand, they state that Pkm silencing leads to a reduced CalexA signalling; however, when looking at Fig. 6D and E, we see a slight increase in the GFP signal (Fig. 6D) and the quantification is similar to control levels (Fig. 6E). The authors should verify this result and interpret accordingly.

We want to thank the reviewer for this comment, we agree and we have redesigned the presentation of these data. The intensity of GFP signal between control and sHsps upregulation is significant, as well as Hsps upregulation and pkm downregulation. However, the differences between control samples and pkm downregulation samples is not statistically significant, even though there is a tendency. We have now included further statistical analysis in the figure and discussed these results in the text thoroughly.

Minor comments:

1. The figures are erroneously numbered: Fig.1 is labelled as Fig. 5; Fig. 2 as Fig.6; Fig. 3 as Fig.1; Fig. 4 as Fig.2; Fig. 5 as Fig.3; Fig. 6 as Fig.4. This is unpleasant!

We apologize for these errors originated by the uploading process in biorXiv; we have now revised and corrected all of them.

2. In Fig. 1C and D, one of the proteins represented in the interactome diagrams is labelled CG143755, while in the text the authors refer to it as CG43755. Furthermore, the authors refer to a protein named CG1561, while in these diagrams it is labelled as Pmk. The authors should already state the new name of this protein in the text or change the labels in the diagrams.

We have corrected all the names and included Pkm in the text referring to CG1561.

3. In Fig. 2A and B, the figure labelling doesn’t match the figure legend. In A, we see the label for sHSP23 and in B for sHSP26 while in the legend, the authors state the contrary.

We have corrected the figure legend.

4. In Fig. 4C-E, there is difference between the figure labelling and the legend description. Fig. 4C shows the quantification of the blot for both sHSP23 and sHSP26 while in the legend they are separated as C and D. Consequently, Fig. 4D is described as 4E in the legend, and so on. Furthermore, the summary table illustrated in Fig. 4D doesn’t bring much information and could be removed.

We have corrected the figure and the legend accordingly

---

## [Decision Letter · Decision Letter 1]

29 Apr 2020

PONE-D-20-03396R1

Small heat shock proteins determine synapse number and neuronal activity during development

PLOS ONE

Dear Casas-Tinto,

Thank you for submitting your manuscript to PLOS ONE. After careful consideration, we feel that it almost acceptable for publication in PLOS ONE’s. The reviewer still has one suggestion that I would aks you to consider. Therefore, we invite you to submit a revised version of the manuscript that addresses this remaining point.

If done, I will immediately proceed and accept your MS.

We would appreciate receiving your revised manuscript by Jun 13 2020 11:59PM. To enhance the reproducibility of your results, we recommend that if applicable you deposit your laboratory protocols in protocols.io, where a protocol can be assigned its own identifier (DOI) such that it can be cited independently in the future. For instructions see: http://journals.plos.org/plosone/s/submission-guidelines#loc-laboratory-protocols

We look forward to receiving your revised manuscript.

Kind regards,

Harm H Kampinga

Academic Editor

PLOS ONE

Reviewers' comments:

Reviewer's Responses to Questions

**Comments to the Author**

1. If the authors have adequately addressed your comments raised in a previous round of review and you feel that this manuscript is now acceptable for publication, you may indicate that here to bypass the “Comments to the Author” section, enter your conflict of interest statement in the “Confidential to Editor” section, and submit your "Accept" recommendation.

Reviewer #1: (No Response)

2. Is the manuscript technically sound, and do the data support the conclusions?

Reviewer #1: Yes

3. Has the statistical analysis been performed appropriately and rigorously? 

Reviewer #1: Yes

4. Have the authors made all data underlying the findings in their manuscript fully available?

Reviewer #1: Yes

5. Is the manuscript presented in an intelligible fashion and written in standard English?

Reviewer #1: Yes

6. Review Comments to the Author

Reviewer #1: The authors have answered to all my comments. I still have one more suggestion concerning Figure 2G. Since the authors aren't sure about the 2 bands of sHsp26, I would add an arrow pointing to the band level that is at the right MW beside the gel and maybe add an * at the second band level, stating in the figure legends that the origin of this second band is still unknown.

7. PLOS authors have the option to publish the peer review history of their article (what does this mean?). If published, this will include your full peer review and any attached files.

Reviewer #1: No

---

## [Author Response · Author response to Decision Letter 1]

30 Apr 2020

We want to thank the reviewer for all his effort to improve our manuscript. We have included an arrow and an asterisk in Fig2G to indicate HSP26 and the unknown band

---

## [Editor Report · Decision Letter 2]

1 May 2020

Small heat shock proteins determine synapse number and neuronal activity during development

PONE-D-20-03396R2

Dear Dr. Casas-Tinto,

We are pleased to inform you that your manuscript has been judged scientifically suitable for publication and will be formally accepted for publication once it complies with all outstanding technical requirements.

With kind regards,

Harm H Kampinga

Academic Editor

PLOS ONE
---

## [Editor Report · Acceptance letter]

8 May 2020

PONE-D-20-03396R2 

Small heat shock proteins determine synapse number and neuronal activity during development 

Dear Dr. Casas-Tinto:

I am pleased to inform you that your manuscript has been deemed suitable for publication in PLOS ONE. Congratulations! Your manuscript is now with our production department. 

With kind regards,

on behalf of

Dr. Harm H Kampinga 

Academic Editor

PLOS ONE